# Patient Perspectives on Coordinated Care: Preliminary Results from the Implementation Stage Using Patient-Reported Experience Measures (PREMs)

**DOI:** 10.3390/healthcare13091026

**Published:** 2025-04-29

**Authors:** Beata Wieczorek-Wójcik, Anna Justyna Milewska, Dorota Kilańska, Aneta Kulma-Pytlak, Peter Iltchev, Aleksandra Gaworska-Krzemińska, Remigiusz Kozlowski

**Affiliations:** 1Department of Nursing and Medical Rescue, Pomeranian University in Slupsk, Westerplatte Street 64, 76-200 Słupsk, Poland; bwwojcik@op.pl; 2Department of Biostatistics and Medical Informatics, Medical University of Białystok, 15-295 Białystok, Poland; anna.milewska@umb.edu.pl; 3Division of Community Nursing & Health Promotion, Medical University of Gdańsk, 80-210 Gdańsk, Poland; 4Department of Coordinated Care, Medical University of Łódź, 91-419 Łódź, Poland; 5Salus Health Center-Primary Health Care Clinic, Szafranka, 76-200 Słupsk, Poland; anetakulma@gmail.com; 6Department of Management and Logistics in Healthcare, Medical University of Łódź, 91-419 Łódź, Poland; petre.iltchev@umed.lodz.pl (P.I.); remigiusz.kozlowski@umed.lodz.pl (R.K.); 7Division of Nursing Management, Medical University of Gdańsk, 80-210 Gdańsk, Poland; aleksandra.gaworska-krzeminska@gumed.edu.pl

**Keywords:** coordinated-care, Health Services Accessibility, evaluation, patient satisfaction, VBHC value-based healthcare

## Abstract

**Background and Objectives:** Integrated health services are health services that are managed and delivered in a way that ensures patients receive a continuum of health promotion, disease prevention, diagnosis, treatment, disease management, rehabilitation, and palliative care services at different levels and sites of care within the health system, and according to their needs, throughout their life course. Assessing the effectiveness of their implementation, the perspective of the process participant—the patient—is examined. There are three main types of patient-reported measures: PROM, PREM and HLS. PREM (patient-reported experience measure) is a tool that allows the objective measurement of the patient’s experience related to healthcare services, for instance, the timeliness of visits or receiving recommendations. The aim of this study was to evaluate the coordinated care experienced by patients (PREMs) before and after the introduction of coordinated care, using the JOP-POP tool as a key measure. **Materials and Methods:** This longitudinal study was conducted in two stages. The first stage concerned the joining of the coordinated care program by the entity in which the study was conducted; the study was repeated six months after joining coordinated care (CC). At each stage of the study, the study group included 40 patients. The Shapiro–Wilk test was used to verify the normality of the distribution of quantitative variables. For statistical analysis, the Wilcoxon test for paired samples was used to compare two ordinal dependent variables. For independent variables, the Mann–Whitney and the Kruskal–Wallis ANOVA by ranks tests were used, with a post hoc test of multiple comparisons of mean ranks. **Results:** A statistically significant relationship (*p* = 0.00157) was observed between the number of chronic diseases and health status assessment before inclusion in coordinated care. The patients’ responses showed statistically significant improvement 6 months after the introduction of coordinated care (CC). The improvement in assessment was related to the time physicians spent with patients. The greatest improvement over 6 months was achieved in coordination of care and the smallest improvement was noted in the approach to the patient. **Conclusions:** The JOP-POP tool may be useful in future studies to assess patients’ experiences with implementing coordinated care.

## 1. Introduction

Integrating healthcare and social services continues to be a global challenge for many health systems. Reports of limited access to healthcare services, prolonged waiting times, high costs, a poor quality of care, and medical errors appear almost daily in the media [1]. The early reformers of the system in the USA and pioneers of value-based healthcare (VBHC) have observed that medical services are often limited or rationed, many patients receive substandard care, and the incidence of preventable medical errors remains high. There are significant and inexplicable differences in cost and quality between suppliers and across geographical regions. These results are unacceptable in healthcare, as health and life are at stake [2]. In addition, the US saw a significant increase in health spending, with a tendency to consume 100% of GDP by 2040 [3]. According to numerous strategic reports, hospitals account for from 40% to over 70% of total healthcare expenditure, with the highest percentages observed in former communist countries [4,5]. Hence, the move towards a new model became obvious.

In Poland, research results by the Center for Social Opinion Research (CBOS) from 2023 point to a low percentage of patients satisfied with healthcare, which was 27%, indicating that compared to 2021, there was a decrease in the number of people who rated healthcare positively (by 2 points), and there was an increase in the number of people who perceived it negatively (by 4 points) [6]. In 2016, 21% of patients were satisfied with their care (prior to the launch of the pilot program of care POZ PLUS coordinated by the Polish National Health Fund). In contrast, according to the System of Health Accounts (SHA) [7], the proportion of medical expenditure within hospital health expenditure has remained at 30–31% since 2013—approximately 10% lower than the overall expenditure on hospitals as institutions. The lowest levels of such expenditure are observed in Portugal and The Netherlands [8].

For healthcare to deliver value to patients, the nature of competition within the sector must change. It is alarming that payers, health policymakers, providers, doctors, and others in the system debate irrelevant issues rather than discuss the critical changes in the right place at the right time. System participants focus on dividing value rather than creating it—and, in some cases, they even diminish it. They shift costs between entities, restrict access to care, stifle innovation, and collect data without providing meaningful benefits to patients. This zero-sum competition must be replaced with a focus on competition in preventing, diagnosing, and treating individual health conditions [2]. Thus, a new approach to care is proposed—value-based healthcare (VBHC). It is a strategy for achieving better outcomes by considering the factors most important to patients, while optimizing the cost of services [9,10]. In 2014, the European Commission urged its member states to implement a method of reimbursement of costs based on the price–quality ratio [11]. One of the six elements of Porter’s VBHC is the integration of care through its coordination [9]. Integrated health services are health services that are managed and delivered in a way that ensures patients receive a continuum of health promotion, disease prevention, diagnosis, treatment, disease management, rehabilitation, and palliative care services at different levels and sites of care within the health system, and according to their needs, throughout their life course [12]. A systematic review identified over 40 definitions of the term “care coordination” [13]. For the purposes of this publication, the following definition resulting from a systematic review has been adopted [14,15].


*“Care coordination is the deliberate organization of patient care activities between two or more participants (including the patient) involved in a patient’s care to facilitate the appropriate delivery of health care services. Organizing care involves the marshaling of personnel and other resources needed to carry out all required patient care activities and is often managed by the exchange of information among participants responsible for different aspects of care”.*
[13]

The Agency for Healthcare Research and Quality defines care coordination as “the deliberate organization of patient care activities between two or more participants (including the patient) involved in a patient’s care to facilitate the appropriate delivery of healthcare services” [16].

Care coordination and care integration are inversely related, because as services become more integrated, the need for coordination decreases [14]. In an organization, the development of care coordination—and consequently, continuity of care—seems more likely to be favored than a system of care networks [15].

A people-centered perception of care (adopting the perspective of the individual) is the pillar of coordinated care. People-centered healthcare services involve a caring approach that deliberately considers the perspectives of individuals, families, and communities, viewing them as active participants and beneficiaries of healthcare systems that address their needs and preferences in a compassionate and holistic manner. People-centered care requires that people have health literacy competencies and the support they need to make decisions and participate in their own care. It is organized around people’s health needs and expectations, not diseases [12,17]. Therefore, in assessing the effectiveness of its implementation, the perspective of the process participant—the patient—is examined. As per the recommendations of the Polish Agency for Health Technology Assessment and Tariff System, the Euro–Quality of Life Questionnaire (EQ-5D, version EQ5D3L or EQ5D-5L) is the preferred instrument for measuring quality of life in adults, as its widespread use ensures the highest comparability of the results [18]. The main challenge facing the Polish healthcare system—and indeed other healthcare systems—is achieving value added (VA) from continuously rising expenditures. To this end, it is necessary to define the system’s objectives and consistently monitor their progress [8], considering the patient’s perspective.

The nominal funding of services provided under the Polish National Health Fund has increased since 2016 by more than 130%. This represents a real growth of over 50%, and after accounting for the dynamics of nominal GDP growth, it stands at 16%. In contrast, the number of services performed increased by only 6.3%. What draws attention is the low growth in the number of outpatient specialized care services, which increased by only 0.4%. What remains a challenge is ensuring that the increase in healthcare funding effectively translates into improved patient access [19].

Confronted with dissatisfaction with care and the expectations of system stakeholders and medical professionals, who called for increased healthcare spending, especially considering rising GDP consumption, the Polish Ministry of Health and the National Health Fund initiated a discussion on coordinated care in 2013. This was in response to the fragmentation of services and aimed at improving the efficiency and effectiveness of the healthcare system. EU funding was granted for developing and piloting a coordinated system of services within the primary healthcare framework in Poland. The preparation, testing, and implementation of coordinated care organization into the healthcare system—“Stage II Pilot Phase—Model POZ PLUS” was carried out by the Polish National Health Fund from 1 October 2017 until 31 March 2022 [20]. The assessment of the project’s effects involved using quality measures evaluated from the patient’s perspective. There are three main types of patient-reported measures: PROM, PREM, and HLS [21]. The PROM measure (patient-reported outcome measure) was used to assess the patient’s health, which covered physical condition, well-being, and mental functioning and included physical and mental difficulties associated with the disease, symptoms, social functioning, and adherence to therapeutic procedures. PREM (patient-reported experience measure) is a tool used to assess patient’s experience related to healthcare services, for instance, the timeliness of visits or receiving recommendations. The evaluation criteria include waiting times for services, accessibility, and ease of use of healthcare services, patient and provider involvement in treatment decisions, knowledge of treatment plans and diagnostic and therapeutic pathways, quality of communication, support in managing chronic illness, and the likelihood of recommending the service to family and friends [22,23]. The third measure, HLS (health literacy survey), describes the patient’s skills, knowledge, and self-confidence in the self-management and understanding of their own health. It is used to evaluate the patient’s health literacy based on their answers to questions about their illness, beliefs, self-confidence, or independence. Health literacy is the capacity of an individual to make informed health decisions in everyday life—at home and at work. It also encompasses healthcare, the marketplace, and health policy [24,25].

The PREM questionnaire used in the pilot POZ PLUS Project in Poland proved to be more demanding in terms of cultural and linguistic adaptation than the PROM questionnaire. With a limited number of PREM questionnaires available in the world, only one of them was suitable for adaptation in the context of the pilot POZ PLUS Project. It included an assessment of the coordination of primary care services with specialized care and hospital care and an evaluation of patient’s social support and their general well-being. The questionnaire for the pilot POZ PLUS Project in Poland was validated following the guidelines of the World Health Organization [26].

At the end of the pilot, a model was implemented in which the coordinator plays a central role in organizing coordinated care within the primary care framework, collaborating with the patient and other medical team members while primarily handling administrative tasks. Thus, the coordinator acts as a navigator, organizing and monitoring the diagnostic and therapeutic process while responding to emergency situations. The role also encompasses patient support and is typically performed by a nurse or a non-medical staff member, such as a person responsible for patient registration [26,27]. The patient receives an individualized medical procedure plan in which primary care professionals carry out interventions appropriate to their level of competence. Educational tasks are usually assigned to the nurse.

The aim of this study was to evaluate the coordinated care as experienced by patients (PREM) before and after the introduction of coordinated care, using the JOP-POP (pol. *Jakość Obsługi Pacjenta—PODaż*—Patient Service Quality—Supply) tool as a key measure [28].

## 2. Materials and Methods

This study used the JOP-POP questionnaire by Rudawska (Appendix A). In Rudawska’s article, its reliability and internal consistency were confirmed by the Cronbach’s alpha (α) test, which was 0.891 for the entire JOP-POP, and for each of the seven areas, it was in the range of 0.884 to 0.896 [29]; JOP-POP is a set of highly correlated statements that allows for the assessment of care from the perspective of patient’s experience (PREMs). It includes 29 statements grouped into 7 areas: *A. patient’s participation in the treatment process; B. patient support by medical personne; and in decision-making; C. continuity of care; D. coordination of care; E. solving patients’ problems; F. availability and flexibility of care; G. approach to the patient*. A five-step Likert scale (1–5) was used to indicate the frequency of the studied items; 1 corresponded to never, 2—rarely, 3—sometimes, 4—almost always, 5—always. The sum of the points obtained is a summary indicator of the patient’s assessment of care. Additionally, it is important to note that the JOP POP questionnaire was validated on a sample of 320 patients in Poland. The sample size was determined using the minimum sample size formula for the structure index, with a confidence interval of 0.95 and a maximum allowable estimation error of 0.05. Patients were qualified based on their willingness to participate. However, the value of Rudawska’s questionnaire lies in the detailed description of the characteristics of the tested variables, which are not always available in other tools [29].

This longitudinal study was conducted in two stages. The first stage concerned the joining of the coordinated care program by the entity in which the study was conducted (signing an agreement with the National Health Fund); the study was repeated six months after joining coordinated care (CC). The study was conducted from 1 June 2023 to 31 December 2023 in a primary care facility, which is a part of a Healthcare Center, *Centrum Zdrowia Salus in Słupsk*, Poland.

The inclusion criteria required consent to participate in the study and enrollment of the patient in coordinated care within the framework of primary health care. At each stage of the study, the study group included 40 patients. The respondents were informed about the subject and purpose of the study; their anonymity and voluntary participation in the study were ensured. The respondents were also asked to provide demographic data (age, gender, education, place of residence) and describe their health status (number of chronic diseases, subjective well-being). Additionally, they indicated the path of coordinated care for which they were qualified (diabetology, cardiology, pulmonology, or nephrology).

The data were recorded in an Excel form. The Shapiro–Wilk test was used to verify the normality of the distribution of quantitative variables. The variables used in this study did not have a normal distribution. For statistical analysis, the Wilcoxon test for paired samples was used to compare two ordinal dependent variables. For independent variables, the Mann–Whitney and Kruskal–Wallis ANOVA by ranks tests were used, with the post hoc test of multiple comparisons of mean ranks. Correspondence analysis was used to present the relationship between the categorized variables. Pearson’s chi-square test was used to evaluate this relationship. The results were considered statistically significant at *p* < 0.05. Statistica 13.3 was used in the calculations.

This is a preliminary study focusing on a single facility and uses a relatively small sample, as coordinated care is yet to become common. Since no random recruitment was planned, the results have limitations, and, as it is an observational study, it may be subject to confounding factors. There is no available ICT system available to analyze structured variables.

## 3. Results

There were twice as many women as men in the study group (Table 1). The majority of respondents were over the age of 50, and most respondents were the residents of cities. Half of the respondents had completed higher education and approximately 25% had completed secondary education. The remaining respondents had primary or vocational education, comprising nearly 25% of the group. The most common pathway was cardiology, followed by endocrinology and diabetology.

A statistically significant relationship (*p* = 0.00157) was observed between the number of chronic diseases and health status assessment before inclusion in coordinated care (before CC). The analysis of the correspondence shows three distinct foci: people with 1 chronic disease assess their status as rather good or very good, people with 2–3 diseases assess their health status as average, and people with more than 3 diseases assess their health status as poor (Figure 1A). There was no statistically significant association between the number of chronic diseases and self-assessment after 6 months of coordinated care (after CC) (*p* = 0.11188).

A comparative analysis of health status before and after the introduction of coordinated care indicated a statistically significant improvement (*p* = 0.000089) (Figure 1B).

Table 2 shows the areas of assessment of patients’ responses before and 6 months after the introduction of coordinated care. There was a statistically significant improvement in all areas.

Table 3 presents information on the statistical significance of the study results for individual findings.

### 3.1. Patient’s Participation in the Treatment Process (A)

The patients’ responses showed a statistically significant increase 6 months after the introduction of coordinated care (CC) across the whole patient participation area (*p* = 0.0015; Table 2). Two of the five tasks performed in this area showed a significant difference in the assessment of patients before and after introducing coordinated care. The increase in assessment values was related to the time physicians spent with patients (Q2: *p* = 0.0431; Figure 2A) and the physician’s encouragement for the patient to share their health problems (Q5: *p* = 0.0180; Figure 2B).

### 3.2. Patient Support in Decision-Making (B)

Patients’ responses within the area of patient support by healthcare professionals in decision-making showed a statistically significant improvement in the assessment of patient support 6 months after the introduction of coordinated care (*p* = 0.0010; Table 2).

Two of the four tasks performed in this area showed a significant difference in the assessment of patients before and after the introduction of coordinated care. The improvement was related to the assessment of support in changes in lifestyle to better manage chronic disease (Q7: *p* = 0.0180; Figure 3A) and the availability of information materials explaining how to manage chronic disease (Q8: *p* = 0.0007; Figure 3B).

### 3.3. Continuity of Care (C)

Patients’ responses showed a statistically significant increase 6 months after the introduction of coordinated care across the continuity of care area (*p* = 0.0002; Table 2).

Three of the four tasks performed in this area showed a significant difference in the assessment of patients before and after the introduction of coordinated care. Improvement was noted in the criteria related to the ability to continue specialized treatment without having to make appointments with a general practitioner to obtain a referral (Q11: *p* = 0.0218; Figure 4A), obtaining information via the general practitioner about recommendations from other doctors, types of medication prescribed and treatments performed in other medical settings (Q12: *p* = 0.0431; Figure 4B), and regularly performed diagnostic tests and systematic follow-up visits resulting from chronic disease (Q13: *p* = 0.0033; Figure 4C).

### 3.4. Coordination of Care (D)

Patients’ responses within the care coordination area showed a statistically significant increase 6 months after the introduction of coordinated care (*p* = 0.0001; Table 2).

Three of the four tasks performed in this area showed a significant difference in the assessment of patients before and after the introduction of coordinated care. The improvement in the assessment concerned the appointment of a person from the medical staff (doctor, nurse) to organize care in the healthcare system (Q14: *p* = 0.0002; Figure 5A), a patient’s physician being able to consult other physicians about the patient’s health (Q16: *p* = 0.0117; Figure 5B), and providing comprehensive healthcare related to the chronic disease of the patient, who can attend many different medical consultations in the same medical institution, e.g., with a dietitian or a physical therapist (Q17: *p* = 0.0032; Figure 5C).

### 3.5. Solving the Patient’s Problems (E)

Patients’ responses across the patient problem-solving area showed a statistically significant increase 6 months after the introduction of coordinated care (*p* = 0.0001; Table 2).

Two of the five tasks performed in this area showed a significant difference in the assessment of patients before and after the introduction of coordinated care. The improvement was noted for the variable of a physician’s explanation of treatment objectives, treatment process, and expected outcomes (Q18: *p* = 0.0180; Figure 6A) and instructions on how to act in the event of a sudden deterioration in health or exacerbation of signs and symptoms of illness provided by the assigned physician or another person from the medical staff (Q20: *p* = 0.0110; Figure 6B).

### 3.6. Flexibility and Availability of Care

Patients’ responses showed a statistically significant improvement in the availability and flexibility of care 6 months after the introduction of coordinated care (*p* = 0.0069; Table 2).

One of the three tasks performed in this area showed a significant difference in the assessment of patients before and after the introduction of coordinated care (before and after CC, respectively). The improvement in the assessment was related to patients’ ability to choose a convenient date for an appointment (Q23: *p* = 0.0071; Figure 7).

### 3.7. Approach to the Patient

Patients’ responses showed a statistically significant improvement in the patient approach area 6 months after the introduction of coordinated care (*p* = 0.0409; Table 2).

However, none of the four tasks performed in this area of tasks showed a significant difference in the assessment of patients before and after the introduction of coordinated care.

### 3.8. Summary Evaluation of the Questionnaire Results

The total score of the entire questionnaire before the introduction of coordinated care was 115 points (median), and it increased significantly, statistically, after 6 months of care (*p* < 0.0001) to 122 points (median). The overall assessment of patient satisfaction did not differ significantly, statistically, between care pathways, gender, place of residence, education, age, and perceived health status either before or after the introduction of coordinated care (*p* > 0.05; Figure 8).

### 3.9. Analysis Comparing Improvements in Each Area

To assess which areas had the strongest and the weakest improvement, we calculated an indicator, which we called the measure of improvement (the difference between the assessment of the area after 6 months and the assessment of the area before the introduction of coordinated care). Descriptive statistics for improvement measures are presented in Table 4. To clearly organize the areas for improvement and create a ranking of the areas, the average sums of the ranks were calculated (the median values were insufficient).

The Kruskal–Wallis test comparing the measures of improvement showed statistically significant differences in improvement between the study areas (*p* = 0.0012). In contrast, the post hoc test showed that the difference between “Area D” and “Area G” was statistically significant (*p* = 0.0076).

The greatest improvement over 6 months was achieved in coordination of care (Area D), and the smallest improvement was noted in the approach to the patient (Area G) (Figure 9).

## 4. Discussion

Coordination of care helps patients, doctors, and other service providers to deal with complex medical conditions within the framework of primary healthcare. The goal of care coordination is to promote communication and continuity of care between providers, specializations, and medical systems, and the goal is to reduce healthcare costs and improve clinical outcomes. Coordinated primary health care was implemented in Poland on 01 October 2022. This is the biggest change in healthcare in decades. It is based on a Patient-Centered Model of Care, and it not only measures the effectiveness of implemented procedures but also focuses on health promotion and prevention. Almost half of the population can access at least one coordination pathway two years after the introduction of coordinated care. At least 38% of primary healthcare institutions have signed an agreement with the National Health Fund for at least one coordination pathway. The results of a study conducted by the “My Pacjenci” Foundation (a think tank advocating patient involvement and patient education) indicate that over 92% of patients are satisfied with their involvement in diagnosis and treatment decisions based on their physician’s treatment plan. More than 93% confirm that the diagnosis and treatment are proceeding according to the plan. These results suggest that communication between doctors and patients is effective and transparent, and that healthcare delivery is well organized [30]. Quite to the contrary, in a survey by the Polish Center for Public Opinion Research, most respondents indicated negative opinions in all assessed areas [6]. Integrated care requires an approach that is both interprofessional and interdisciplinary. However, several obstacles can hinder its implementation, including rigid organizational structures, siloed specialization, performance metrics focused solely on individual outcomes in limited areas, and the underutilization of nurses’ full scope of competence, unlike in the WHO-recommended Chronic Care Model [31]. The model was initiated in the United States in 1997 and has been validated through research. Its success depends on the availability of all its core components; missing even a single element can prevent the achievement of the intended health outcomes [32,33].

Our survey indicated improved perceptions across all the variables identified in the questionnaire. Statistically significant improvement was observed in all seven areas (A–G). Despite an overall statistically significant improvement, more than half of the individual survey items showed improved scores that were not statistically significant. This was the case for patient participation in the treatment process (area A)—3/5 variables; patient support by medical staff—3/5 variables; solving patient’s problems (Area E)—this applies to 3 out of 5 variables; flexibility and availability of care (Area F)—2 out of 3 variables; approach to the patient (Area G)—4 out of 5 variables. This can be considered a starting point for the further consideration of implementing coordinated care.

Coordinated care significantly enhances the management of hypertension and diabetes, reduces emergency room admissions, and improves patient well-being [34,35]. The discussion focused on areas of intervention that showed statistically significant results.

### 4.1. Patient’s Participation in the Treatment Process (A)

The results of this study indicate an improvement in patient participation in the treatment process, which may be due to focusing on the patient and spending more time with them, since coordinated care appointments are usually made at the end of the appointment schedule for a given day. Hence, there is no pressure caused by the queue of other waiting patients. A statistically significant improvement was observed for two variables (QA_2 and QA_5). Patients reported improvements in their treatment plan, information about diagnostic tests, and lifestyle changes [36]. A meta-analysis of the literature and other systematic reviews indicate that involving a patient in care planning personalizes care by recognizing its strengths and the patient’s hopes and preferences based on individual beliefs, values, cultural contexts, and personal situations [37]. Patient involvement in care has become crucial to the success of healthcare reforms in many countries around the world [38]. The concept of patient-centered care started replacing the paternalistic model of care as early as in the 1990s [39]. Decades of research have shown that patient involvement is key to this approach [34,40,41,42,43,44]. This is particularly true for patient-centered methods, such as motivational interviewing [45], and the increasingly accessible IT tools that enhance patient involvement [46,47,48].

### 4.2. Continuity of Care (C)

According to the Polish Center for Public Opinion Research, in 2023, access to specialist doctors received one of the lowest ratings by patients, with 84% of reviews being negative. However, through the coordination of care in the study clinic, the largest group of stakeholders in the study included patients who—in most cases—could continue to receive specialized treatment without the need for a referral. An increase of 7% was noted in this group. This change primarily stems from the expanded range of services now available within each diagnostic and therapeutic pathway. Respondents accessed specialized services for the comprehensive diagnosis and treatment of chronic diseases with stabilized progression. Similar opinions were obtained for coordinated care in other countries [49].

Technology is playing an increasingly vital role in bridging gaps in healthcare co-ordination by facilitating communication and telemedicine, a trend significantly accelerated by the COVID-19 pandemic [50]. This technology facilitates communication among multiple providers via electronic medical records and enhances interactions between patients and providers through patients’ portals and similar tools [51,52]. This study shows that physicians can access information about consultations provided by other healthcare professionals through electronic documentation and the search function for medical events on gabinet.gov.pl. Most patients provide up-to-date information from other specialists, which is subsequently added to their medical records.

The steps within the course of treatment have been organized through the establishment of an individualized medical procedure plan, and the coordinator supervises their proper implementation.

### 4.3. Coordination of Care (D)

A statistically significant improvement was observed in the appointment of a coordinator for patients and in consultations between the attending physician and other doctors. Patients participating in the study were introduced to the coordinator in person. Each patient was given a contact number to discuss issues related to the implementation of the care plan directly. All appointments, tests, and examinations included in IPOM were scheduled only by the coordinator. Other studies have also confirmed satisfaction with the coordinator in the discussed model of care. Patients appreciate the help of care coordinators in moving through the care system, engaging patients in self-care, and educating them [30,49]. Also, in the pilot phase of POZ PLUS, respondents indicated the effectiveness of cooperation with the coordinator [27]. When interacting with healthcare providers, patients emphasize the following as the most significant: (a) when they are viewed from a holistic perspective, i.e., they are viewed holistically rather than solely in the context of their illness, (b) their health experiences are acknowledged as valid, (c) they are recognized as experts in understanding their own needs, (d) they are valued as experts on themselves and their lives, and (e) they collaborate with healthcare providers to navigate the system and access necessary care and services [53]. Nurse-coordinated care has been linked to positive patient experiences and improvements in symptom management and lifestyle outcomes [54]. The impact of interdisciplinary cooperation (involving doctors, nurses, dietitians, and physiotherapists) should also be emphasized, particularly in terms of treatment comprehensiveness, a reduction in complications, and a shorter time to diagnosis and treatment period length [13,55]. Notably, the role of the care coordinator, often a nurse, has clinical value not only in facilitating access and communication but also in improving patient outcomes. Evidence supports nurse-led services that enhance symptom management, promote healthier lifestyles, and ensure continuity across care settings [53]. In our study, this role contributed to better patient navigation and follow-up, which may translate into improved disease control. Integration with other professionals (e.g., dietitians, physiotherapists) further enhances the comprehensiveness of care, reflecting current evidence that interdisciplinary approaches are essential to effective chronic disease management [54].

A key strategy for reducing care fragmentation and addressing it effectively is to deliver care proactively and systematically. Comprehensive care has many definitions [55], and it depends on factors related to the patient and the care process, such as the number of caregivers involved in patient care. For care to be effective, it should be tailored to the context of the patient and his/her capability and situation. The presented study shows that patients perceived an improvement in the comprehensiveness of care. Other care providers were involved in the process, with their services available at the location where coordinated care was provided. For example, research and consultations, such as dietary advice, were coordinated and carried out within one institution.

Patient communication and education play an important role in coordinated care. The gold standard of practice involves assessing health competence and mental status, as well as applying appropriate conduct models when implementing a change [56]. A statistically significant improvement in the scope of advice and consultation was noted in QB7-8, QE18-20. It should be noted that nurses were involved in this process. The advice included guidance on managing the disease in the event of complications. Research confirms that nurses’ involvement in the education process enhances patients’ self-management of disease, improves efficiency, and reduces costs [37,57].

What is noteworthy is patients’ self-assessment of their health status. Patients with multiple diseases rated their well-being as poorer. Other studies have shown that patients with two or more chronic conditions experienced a larger adjusted mean difference in quality of life compared to those with a single chronic condition [58].

Conventional medical care often fails to meet the needs of chronically ill patients, even within managed and integrated care delivery systems. The literature suggests strategies to improve outcomes in this group of patients. Effective interventions typically focus on one of five key areas: the use of evidence-based interventions and care planning; reorganizing practice systems and provider roles; improving patient self-management support; increased access to specialist care; and improving the availability of clinical information. What remains challenging is organizing these elements into an integrated system of care in the care of patients with chronic diseases [59].

The implementation of coordinated care pathways in primary healthcare settings holds clinical significance beyond patient satisfaction. Studies have demonstrated that such models improve the control of key clinical parameters, particularly among patients with chronic conditions like hypertension or diabetes. For instance, improvements in blood pressure regulation, HbA1c levels, and medication adherence have been reported in similar interventions, contributing to an overall reduction in acute exacerbations and hospitalizations [38]. Our findings, while based on patient-reported experience, align with this broader trend, suggesting a potential clinical benefit that warrants a further outcome-focused evaluation using PROMs (patient-reported outcome measures) [23].

The study has limitations. This preliminary study focuses on a single facility and is based on a relatively small sample, as coordinated care is not common, and there is no ICT system available to analyze structured variables. The aim was to find out whether patient satisfaction following the implementation of coordinated care could be assessed using tools other than those used in the PILOT PLUS survey [26,60]. The JOP POP questionnaire was validated on a sample of patients in Poland and is not widely used for evaluation. However, its value lies in the detailed description of the characteristics of the tested variables, which are not always available in other tools.

While this study employed a validated PREM tool (JOP-POP), future research should complement an experience-based assessment with outcome measures to fully capture the clinical impact of coordinated care. PROMs provide critical insights into changes in patients’ physical, mental, and functional status and have been successfully used in similar care models to monitor improvements in health-related quality of life [22,61]. A combined PREM-PROM approach may thus offer a more robust evaluation of the effectiveness of integrated care programs and should be considered in subsequent, large-scale studies [26,60].

The limited availability of literature also presents a challenge. Most research on coordinated care dates to the 1990s, when such models were being developed and analyzed. In Poland, the system is still in the early stages of transformation, with changes being implemented gradually. Nurses do not yet operate at the full scope of their competencies, and their services are not distinctly defined or reimbursed. As a result, it is difficult to compare the current system with fully developed models that include all key stakeholders. Consequently, conclusions must be drawn with caution and are intended to guide future research directions. Moreover, patient experience measures (PREMs) alone are not sufficient to fully assess the effectiveness of interventions. That is why a parallel analysis of patient-reported outcome measures (PROMs) is also necessary, as PROMs reflect direct health outcomes from the patient’s perspective [23].

## 5. Conclusions

This preliminary study demonstrates that the implementation of coordinated care in primary healthcare settings positively influences the patient experience. Based on patient-reported experience measures (PREMs) collected using the JOP-POP tool, several conclusions can be drawn. The first one concerns the impact on patient experience. Coordinated care was associated with statistically significant improvements across all assessed dimensions of care, including participation in the treatment process, support in decision-making, continuity, coordination, problem-solving, availability, and approach to the patient. The second area is related to the role of care coordinator. The appointment of a designated care coordinator significantly improved perceptions of the organization of care. Coordinators acted as key facilitators, helping patients navigate complex care pathways and accessing multidisciplinary services within a single institution. Furthermore, involving patients in the planning of care fostered greater engagement and self-management. Educational support, particularly from nurses, contributed to developing patients’ health literacy and disease management competencies. Finally, the JOP-POP questionnaire proved useful in capturing nuanced patient experiences related to coordinated care. It may be considered a valuable instrument for future research and quality improvement initiatives in integrated care settings. Further large-scale and multicenter studies are needed to validate these findings and support the wider implementation of coordinated care models in the Polish healthcare system and internationally.

## Figures and Tables

**Figure 1 healthcare-13-01026-f001:**
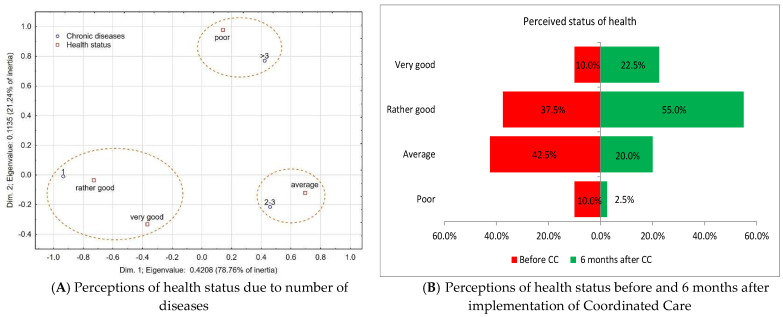
The analysis of the relationship between the number of chronic diseases and the perceived health status of the subjects before and after joining the coordinated healthcare program.

**Figure 2 healthcare-13-01026-f002:**
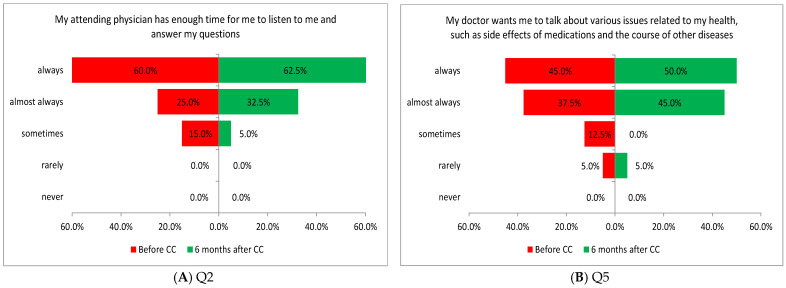
(**A**,**B**) Patients’ opinions about participation in the treatment process.

**Figure 3 healthcare-13-01026-f003:**
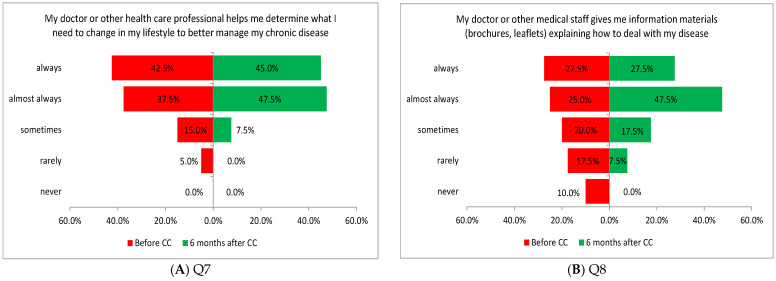
(**A**,**B**) Opinions of patients on the support they receive before and after the introduction of coordinated care (before CC and after CC, respectively).

**Figure 4 healthcare-13-01026-f004:**
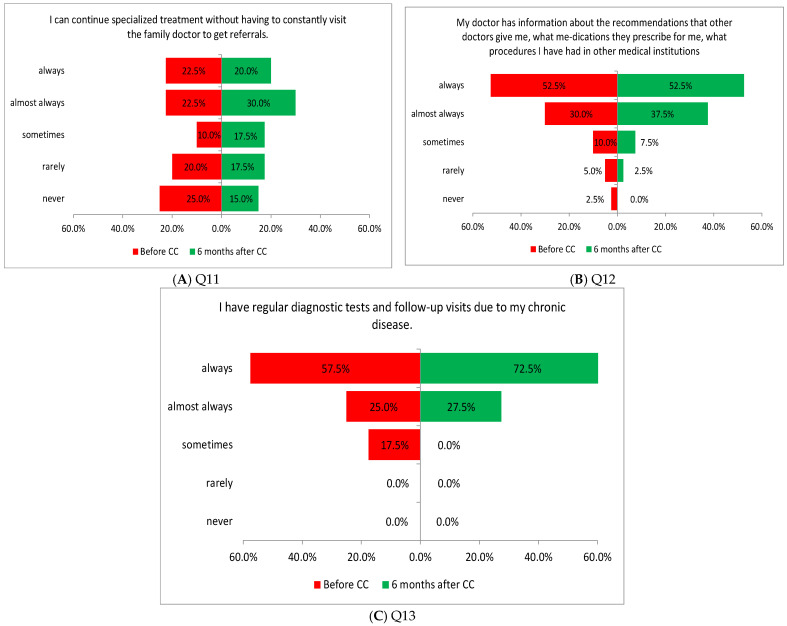
(**A**–**C**) Patients’ opinions on ensuring continuity of care before and after the introduction of coordinated care (before CC and after CC, respectively).

**Figure 5 healthcare-13-01026-f005:**
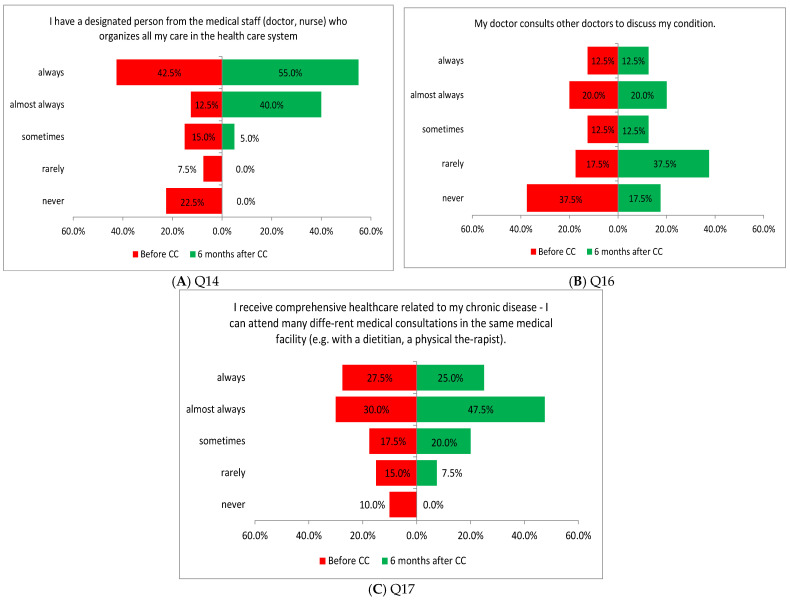
(**A**–**C**) Patients’ opinions on the coordination of care before and after the introduction of coordinated care (before CC and after CC, respectively).

**Figure 6 healthcare-13-01026-f006:**
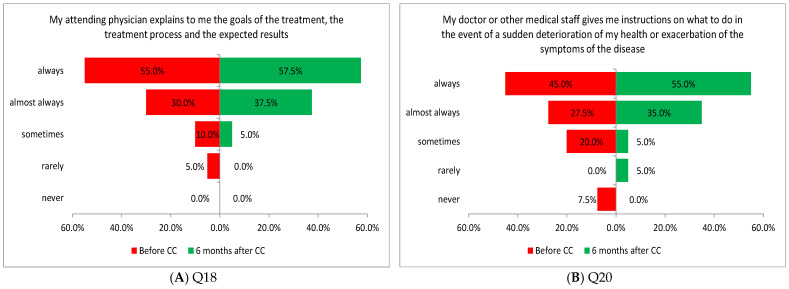
(**A**,**B**) Patient feedback on support in handling disease management challenges.

**Figure 7 healthcare-13-01026-f007:**
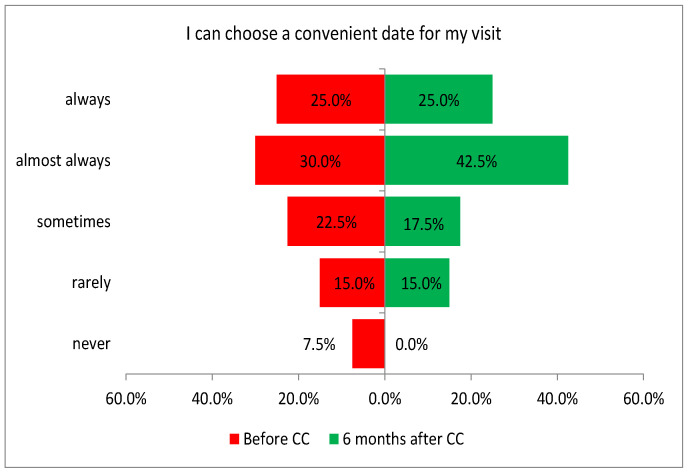
Patient feedback on the availability of care and its flexibility.

**Figure 8 healthcare-13-01026-f008:**
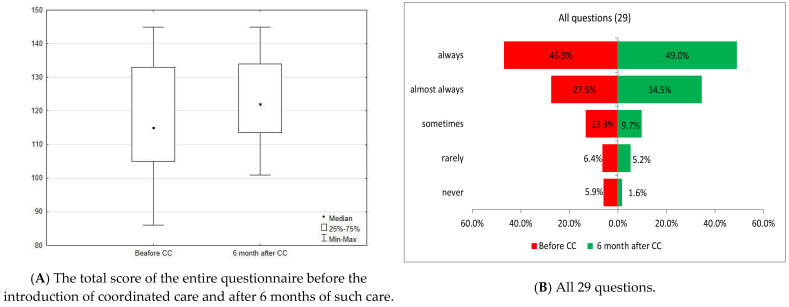
(**A**) Summary assessment before and after the introduction of coordinated care. (**B**) Opinions of patients before and 6 months after the introduction of coordinated care (before and after CC)—summary of 29 variables.

**Figure 9 healthcare-13-01026-f009:**
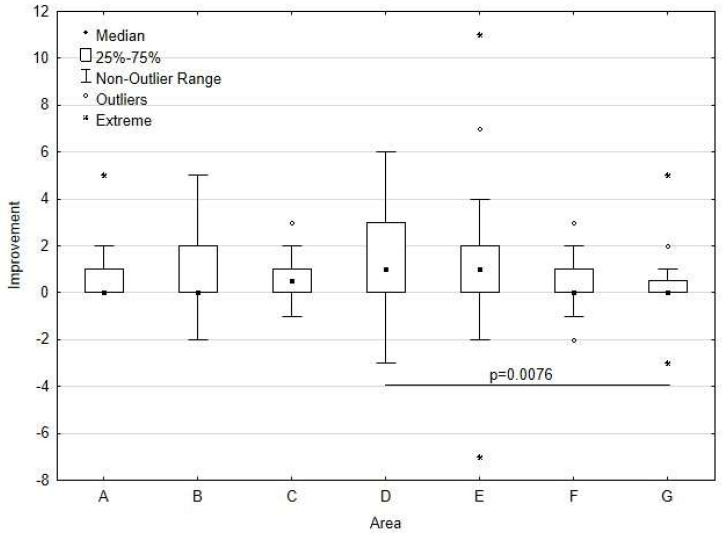
Differences in improvement between the study areas.

**Table 1 healthcare-13-01026-t001:** Characteristics of the study group.

Variable	Feature	N	%
Gender	Female	28	70
	Male	12	30
Age	Up to 35 years	3	7.5
	36–50 years	11	27.5
	51–65 years	13	32.5
	Over 65 years	13	32.5
Place of residence	Village	8	20
	City	32	80
Education	Primary	2	5
	Vocational	7	17.5
	Secondary	11	27.5
	Higher	20	50
Coordinated care pathway	Diabetology	9	22.5
	Endocrinology	14	35
	Cardiology	17	42.5
Number of chronic diseases	1	13	32.5
	2–3	21	52.5
	>3	6	15

**Table 2 healthcare-13-01026-t002:** Patients’ responses in key areas of coordinated care.

Area	Summary Rating “Before” (Median, Quartiles)	Overall Rating “After” (Median, Quartiles)	*p*
A (Q1–Q5)	22.5 (19.5; 24.0)	23.0 (21.0; 24.0)	0.0015
B (Q6–Q9)	18.0 (15.0; 19.0)	18.0 (17.0; 19.0)	0.0010
C (Q10–Q13)	16.5 (14.0; 19.0)	18.0 (15.5; 19.0)	0.0002
D (Q14–Q17)	13.0 (10.0; 17.0)	15.0 (13.0; 17.0)	0.0001
E (Q18–Q22)	21.0 (18.0; 24.0)	21.5 (19.0; 24.0)	0.0001
F (Q23–Q25)	13.0 (11.0; 14.0)	13.0 (11.0; 14.0)	0.0069
G (Q26–29)	17.0 (14.0; 19.0)	17.0 (14.0; 19.0)	0.0409

**Table 3 healthcare-13-01026-t003:** Results of longitudinal studies: statistical significance of findings and individual areas.

Study Area	NoQ	Statements	*p*
A. Patient’s participation in the treatment process	Q1	My doctor appreciates my opinion about the treatment, its course and results.	0.0679
	Q2	My doctor has enough time to listen to me and answer my questions.	0.0431
	Q3	My doctor discusses with me the next steps in the treatment of my disease.	0.0679
	Q4	My doctor gives me the opportunity to choose the method of treatment if there are different methods of therapy (treatment) for my disease.	0.0679
	Q5	My doctor wants me to talk about various issues related to my health, such as the side effects of the medications used and the course of other conditions.	0.0180
B. Patient support in decision-making	Q6	My doctor informs me of additional tests and medical consultations recommended for my chronic illness.	0.1415
	Q7	My doctor or another member of the medical staff helps me determine what needs to be changed in my lifestyle to better manage my chronic illness.	0.0180
	Q8	My doctor or another member of the medical staff gives me information materials (brochures, leaflets) explaining how to deal with my illness.	0.0007
	Q9	I have free access to my medical records.	0.1797
C. Continuity of care	Q10	I have a doctor assigned.	>0.999
	Q11	I have the opportunity to continue specialized treatment without having to constantly visit the family doctor to get referrals.	0.0218
	Q12	My doctor has information about the recommendations that other doctors give me, what medications they prescribe for me, what procedures I have had in other medical institutions	0.0431
	Q13	I have regular diagnostic tests and follow-up visits due to my chronic illness.	0.0033
D. Coordination of care	Q14	I have a designated person from the medical staff (doctor, nurse) who organizes all my care in the healthcare system.	0.0002
	Q15	The doctors who treat me do not contradict each other; I get consistent/the same medical advice from different doctors.	0.5930
	Q16	My doctor consults other doctors to discuss my condition.	0.0117
	Q17	I receive comprehensive healthcare related to my chronic illness—I can attend many different medical consultations at the same medical facility (e.g., with a dietitian or a physical therapist).	0.0032
E. Solving the patient’s problems	Q18	My doctor explains the aims of the treatment, the treatment process, and the expected results.	0.0180
	Q19	My doctor or another member of the clinic staff explains to me what to do next with the treatment, where to go next, how to proceed, who to contact, etc.	0.0756
	Q20	My doctor or another person from the medical staff gives me instructions on how to act in the event of a sudden deterioration in health or exacerbation of the signs and symptoms of the disease.	0.0110
	Q21	I can request a home visit from a doctor or a nurse.	0.1925
	Q22	I can contact the person responsible for my care (community nurse, my doctor) by phone or email.	0.0619
F. Availability and flexibility of care	Q23	I can choose a convenient date for my visit.	0.0071
	Q24	I have access to information about the services offered and the waiting time (e.g., by phone, on the website of the institution).	0.0759
	Q25	The results of my laboratory tests are available on time, i.e., before my next visit to my doctor.	>0.999
G. Approach to the patient	Q26	The staff of the facility respond quickly to my needs; I feel the staff are interested in me.	0.0587
	Q27	In the facility, there is access to information about the patient’s rights, including the possibilities and ways of managing a complaint or contact with a supervisor.	0.1797
	Q28	The facility where I am treated is interested in what patients think about it; it conducts satisfaction research.	0.3613
	Q29	The facility where I am being treated is making changes that I believe will improve patient services.	>0.999

Source: based on own research [29].

**Table 4 healthcare-13-01026-t004:** Ranking of the evaluated areas.

Area	Median	Q1–Q3	Average Sum of Ranks	Ranking
A. Patient’s participation in the treatment process	0.0	0.0–1.0	125.44	V
B. Patient support in decision-making	0.0	0.0–2.0	141.85	Iv
C. Continuity of care	0.5	0.0–1.0	143.16	Iii
D. Coordination of care	1.0	0.0–3.0	175.78	I
E. Solving patient’s problems	1.0	0.0–2.0	161.35	Ii
F. Flexibility and availability of care	0.0	0.0–1.0	124.73	Vi
G. Approach to the patient	0.0	0.0–0.5	111.20	Vii

## Data Availability

The datasets used and/or analyzed during the current study are available from the corresponding author upon reasonable request.

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
