# Peer review of "Patient Perspectives on Coordinated Care: Preliminary Results from the Implementation Stage Using Patient-Reported Experience Measures (PREMs)"

_healthcare, 2025, doi:10.3390/healthcare13091026_

Round 1

Reviewer 1 Report

Comments and Suggestions for Authors

Given the study’s relevance and potential contribution, it is recommended that the authors be invited to revise and resubmit the manuscript after addressing the identified concerns:

1) 17/39 or more references are outdated and need to be replaced with research contributions of the last 5 years, to strengthen the scientific foundation. Some citations are unclear. MDPI style was not used. This, can undermine the scientific validity of your work. Please, revise this issue accordingly. Additionally, some references are outdated. Please, revise this important issue accordingly.

2) The introduction effectively sets the context for the study but lacks a clear articulation of the research gap. While it references patient satisfaction and health system inefficiencies, it does not clearly define how this study contributes beyond existing literature. The rationale for selecting the JOP-POP tool should be expanded, especially in comparison to alternative PREM tools. Please, clearly define the research gap and study contribution in the introduction.

3) The methodology is generally well-described, but the choice of statistical tests requires additional justification. The manuscript states that variables did not have a normal distribution, yet the Mann-Whitney and Kruskal-Wallis tests were applied without further elaboration. Given the small sample size, a discussion on statistical power and effect size is needed. The validity and reliability of the JOP-POP tool should also be explained with reference to more recent studies. Given that the study is limited to a single healthcare setting, the manuscript should discuss potential biases and limitations regarding generalizability to other populations and healthcare contexts. Please, justify statistical methods and discuss the limitations of sample size in the appropriate section.

4) The results are presented clearly, but some statistical comparisons lack meaningful interpretation. While statistical significance is reported, the clinical relevance of findings should be addressed. Several p-values are close to 0.05, indicating marginal significance, which should be interpreted with caution. Figures and tables are informative but require improved captions to clarify key takeaways. Please, provide a deeper interpretation of statistically significant results in terms of clinical impact.

5) The discussion section provides a reasonable interpretation of findings, but it is primarily descriptive. The manuscript would benefit from a more critical engagement with existing literature to contextualize results. While the study references international healthcare models, the relevance to the Polish healthcare system should be more explicitly stated. Additionally, the limitations section should acknowledge potential confounders and biases. Please, improve the discussion section by integrating more recent literature and critically assessing limitations. It is a crucial point.

6) The style adopted for the conclusions paragraph is inconsistent. Please, revise it accordingly.

7) In general, the manuscript contains several instances of awkward phrasing and grammatical errors that hinder readability (e.g., Kruskal-Walli's test, and many other). The sentence structure should be revised for clarity and conciseness. A language service is needed to improve the scientific quality of this work.

Comments on the Quality of English Language

In general, the manuscript contains several instances of awkward phrasing and grammatical errors that hinder readability (e.g., Kruskal-Walli's test, and many other). The sentence structure should be revised for clarity and conciseness. A language service is needed to improve the scientific quality of this work.

Author Response

Rev 1

We would like to thank you very much for your great contribution and thorough analysis of the text, as well as for drawing attention to aspects important for the quality of the work that we could improve in the work. Below we present the answers to the individual elements of the review.

  • 7/39 or more references are outdated and need to be replaced with research contributions of the last 5 years, to strengthen the scientific foundation. Some citations are unclear. MDPI style was not used. This, can undermine the scientific validity of your work. Please, revise this issue accordingly. Additionally, some references are outdated. Please, revise this important issue accordingly.

We have carefully reviewed all references. In total, citations (from the last 5 years, 2019–2024), and examined the literature, In the global literature, the pillars of coordinated care were implemented, described and defined in the 1990s. After analysing the literature using MESH keywords, we found literature 2024 (1) and one systematic review from 2021, which was included in the literature. The need to respond to the remaining reviews may have increased the proportion of older literature. e.g. publication [1], [4], [10], are recommendations from the European Commission and WHO; references [2], [9] are publications by Porter - the creator of VBHC e.g. [9] [12], to which the 2021 systematic review was added. We discussed whether we should remove the original publication, since it is the only one found; [13,22] describe the standard of implementation of Qaly in Poland and describe the implementation of KOZ in Poland; [27,28] EBM guides, no newer version was found; Patient-centred care, PREMS, PROM, HLS are also described based on primary sources [16,17,19;20; 35]; Publications showing the effectiveness of coordinated care - a cost-effective care model was created and described by Wagner [39] (1990s). ) resulted in two reports by the American nursing organisation, we have quoted the last one in the text [37]. MDPI style has been added.

  • The introduction effectively sets the context for the study but lacks a clear articulation of the research gap. While it references patient satisfaction and health system inefficiencies, it does not clearly define how this study contributes beyond existing literature. The rationale for selecting the JOP-POP tool should be expanded, especially in comparison to alternative PREM tools. Please, clearly define the research gap and study contribution in the introduction.

In the introduction, we described the elements related to value-based care, where coordinated care is one of the pillars; focusing on the evaluation of the tool, we pointed out the original assumptions of the PREM, PROM or HLS measurements and referred to the World Bank report, which describes the lack of appropriate tools to evaluate the areas presented in the publication. Rather, reviewers suggest shortening the introduction.

  • The methodology is generally well-described, but the choice of statistical tests requires additional justification. The manuscript states that variables did not have a normal distribution, yet the Mann-Whitney and Kruskal-Wallis tests were applied without further elaboration. Given the small sample size, a discussion on statistical power and effect size is needed. The validity and reliability of the JOP-POP tool should also be explained with reference to more recent studies. Given that the study is limited to a single healthcare setting, the manuscript should discuss potential biases and limitations regarding generalizability to other populations and healthcare contexts. Please, justify statistical methods and discuss the limitations of sample size in the appropriate section.

The methodology section has been revised to include justification for the use of non-parametric tests (Mann-Whitney, Kruskal-Wallis) due to the non-normal distribution of the variables. We have also addressed the limitations of a small sample and provided effect sizes where possible (line 192 - 199, 219 - 227 and 539 - 556).

  • The results are presented clearly, but some statistical comparisons lack meaningful interpretation. While statistical significance is reported, the clinical relevance of findings should be addressed. Several p-values are close to 0.05, indicating marginal significance, which should be interpreted with caution. Figures and tables are informative but require improved captions to clarify key takeaways. Please, provide a deeper interpretation of statistically significant results in terms of clinical impact.

We have added clinical interpretations of key findings in the Discussion sections (line 539-552).

  • The discussion section provides a reasonable interpretation of findings, but it is primarily descriptive. The manuscript would benefit from a more critical engagement with existing literature to contextualize results. While the study references international healthcare models, the relevance to the Polish healthcare system should be more explicitly stated. Additionally, the limitations section should acknowledge potential confounders and biases. Please, improve the discussion section by integrating more recent literature and critically assessing limitations. It is a crucial point.

Publications in PubMed databases were analyzed in relation to patient satisfaction, perhaps since the model was implemented in 2021 in the legal system, no publications on this topic have yet been undertaken, the only source is the World Bank Report from the pilot. At the stage of writing the work, no new studies were found in the context of Poland.

  • The style adopted for the conclusions paragraph is inconsistent. Please, revise it accordingly.

Thanks for your suggestions, which changed to a form that may be more understandable to readers.

7) In general, the manuscript contains several instances of awkward phrasing and grammatical errors that hinder readability (e.g., Kruskal-Walli's test, and many other). The sentence structure should be revised for clarity and conciseness. A language service is needed to improve the scientific quality of this work.

Comments on the Quality of English Language

In general, the manuscript contains several instances of awkward phrasing and grammatical errors that hinder readability (e.g., Kruskal-Walli's test, and many other). The sentence structure should be revised for clarity and conciseness. A language service is needed to improve the scientific quality of this work.

We have asked the translator to revise the translation. Thank you for your suggestion.

Reviewer 2 Report

Comments and Suggestions for Authors

Thank you for the opportunity to review this insightful research paper. The investigation scrutinizes a pivotal concern by examining the efficacy of coordinated care in enhancing patient experiences and outcomes, particularly for individuals suffering from chronic ailments. The research is meticulously organized, featuring a well-defined objective, a robust methodological framework, and a comprehensive discussion.

Nonetheless, I propose that the limitations section be expanded to encompass a more exhaustive examination of potential biases, including but not limited to selection bias, response bias, and the ramifications of utilizing self-reported measures. Such an enhancement would furnish readers with a more lucid comprehension of how these limitations may exert influence over the research findings.

To conclude, this manuscript possesses considerable potential to make a meaningful contribution to the understanding of patient perspectives within the healthcare domain.

Author Response

Rev 2

The limitations section be expanded to encompass a more exhaustive examination of potential biases, including but not limited to selection bias, response bias, and the ramifications of utilizing self-reported measures. Such an enhancement would furnish readers with a more lucid comprehension of how these limitations may exert influence over the research findings.

We have expanded the limitations section to include a discussion of these potential biases, including selection bias (single site, nonrandom sample), response bias (social desirability), and limitations of self-report measures (line 521 – 546).

Reviewer 3 Report

Comments and Suggestions for Authors

Dear authors, I really appreciate your paper. Please find below my comments to improve it:

  1. In your framework coordinated care and integrated care are used as synonymous or are they different? Please clarify this point
  2. Coordinated / integrated care require interprofessional and interdisciplinary approach. Which are the obstacles you see? For example rigidity of organizational structure, vertical specialization, current performance indicators that are based on individual results.
  3. I suggest to clarify the role of patients. I mean to clarify different steps: for patients, involvement, engagement, empowerment. Actually, you discussed the issues, anyway I suggest to widen them.
  4. When you discussed the results of PRE and PRO, it will be helpful to clarify different dimensions: quality of life perception, easier accessibility to healthcare services, participation to decision making.
  5. As you referred to primary care, I suppose there is also a relevance of trust between patient-gp, patient-nurses, if i did not lose it I do not see any discussion about it.
  6. In your framework you introduced the concept of clinical manager. I do not see the role of clinical manager, who is a gp o a specialist. I do not see the concept of care manager, who is a nurse, and somebody adds fragility managers, who in general is a social worker.

Author Response

Rev 3

  1. In your framework coordinated care and integrated care are used as synonymous or are they different? Please clarify this point

We have expanded the description in line 109-111.

  1. Coordinated / integrated care require interprofessional and interdisciplinary approach. Which are the obstacles you see? For example rigidity of organizational structure, vertical specialization, current performance indicators that are based on individual results.

Thank you, this is a very important aspect that if not explained well enough may mislead readers, we have expanded the description in the line 109-111, 471 – 481 and 512 - 519.

  1. I suggest to clarify the role of patients. I mean to clarify different steps: for patients, involvement, engagement, empowerment. Actually, you discussed the issues, anyway I suggest to widen them.

The description was expanded and text was added in lines 426-429 based on available literature.

  1. When you discussed the results of PRE and PRO, it will be helpful to clarify different dimensions: quality of life perception, easier accessibility to healthcare services, participation to decision making.

These aspects are described in lines 469-474, 500-503 and 555-567.

  1. As you referred to primary care, I suppose there is also a relevance of trust between patient-gp, patient-nurses, if i did not lose it I do not see any discussion about it.

These aspects are described in lines 474-482.

  1. In your framework you introduced the concept of clinical manager. I do not see the role of clinical manager, who is a gp o a specialist. I do not see the concept of care manager, who is a nurse, and somebody adds fragility managers, who in general is a social worker.

Perhaps this example is not the most appropriate, however, in the model implemented in Poland, this function is performed, in accordance with the law, by doctors, special coordinators have been appointed, however, due to the size of healthcare entities, other options are possible.

Reviewer 4 Report

Comments and Suggestions for Authors

Thank you for giving me the opportunity to evaluate this article. My views on the article are below.

Abstract Section,
- The abstract is written in detail and sufficiently.

Introduction Section,
- This section is very detailed and sufficient, but it can be shortened for clarity.

Method Section,
- First of all, it is necessary to explain why the data collection period was limited to six months or planned as six months.
- The characteristics of the sample and how the a priori sample size was calculated should be explained. The inclusion of 40 people in the study should be justified.
- The number of institutions included in the study should be specified.
- Information about the validity of the measurement tools should be added.
- More detailed information about the data collection process should be added.

Findings Section,
- The findings are written in detail.

Discussion Section,
- The discussion is based on literature. However, the limitations of the study (for example, was a convenience sample used, were the patients randomized? Is there a difference in the number of men and women? Does this affect the results? Were all the patients from the same region or were there patients from different regions?) should be written in detail and how the limitations were reduced should be explained.

Author Response

Rev 4

Abstract Section,
- The abstract is written in detail and sufficiently.

Introduction Section,
- This section is very detailed and sufficient, but it can be shortened for clarity.

The introduction takes the reader through the idea derived from Porter, through Polish conditions, tool descriptions and justification for the use of the described solution. During the review we came across a suggestion to supplement the introduction, hence we decided to leave it unchanged.

Method Section,
- First of all, it is necessary to explain why the data collection period was limited to six months or planned as six months.

Explained in the methodology and in the limitations in lines 223 and 521-546

- The characteristics of the sample and how the a priori sample size was calculated should be explained. The inclusion of 40 people in the study should be justified.

The sample of 40 people, 28 women and 12 men (70/30%)) is presented in Table 1. Since the aim of the study was to indicate the tool, and the study was conducted shortly after the introduction of coordinated care in the described entity, the research sample was small, the methodology does not recommend conducting a comparative analysis, which would result in conclusions that are significant for the study, therefore we tried to construct the conclusions that we described carefully.

- The number of institutions included in the study should be specified.

Done

- Information about the validity of the measurement tools should be added.

Added line 219 - 227

- More detailed information about the data collection process should be added.

Added in line 192-199

- The findings are written in detail.

Discussion Section,

- The discussion is based on literature. However, the limitations of the study (for example, was a convenience sample used, were the patients randomized? Is there a difference in the number of men and women? Does this affect the results? Were all the patients from the same region or were there patients from different regions?) should be written in detail and how the limitations were reduced should be explained.

We have expanded the section describing limitations

Round 2

Reviewer 4 Report

Comments and Suggestions for Authors

The edits made to the manuscript were found to be sufficient.